# Isolated Atrial Fibrillation, Inflammation and Efficacy of Radiofrequency Ablation: Preliminary Insights Based on a Single-Center Endomyocardial Biopsy Study

**DOI:** 10.3390/jcm12041254

**Published:** 2023-02-04

**Authors:** Roman E. Batalov, Mikhail S. Khlynin, Yulia V. Rogovskaya, Svetlana I. Sazonova, Roman B. Tatarskiy, Nina D. Anfinogenova, Sergey V. Popov

**Affiliations:** 1Cardiology Research Institute, Tomsk National Research Medical Center, Russian Academy of Science, 634012 Tomsk, Russia; 2Arrhythmology Research Department, Federal Almazov Medical Research Centre, 197341 Saint-Petersburg, Russia

**Keywords:** atrial fibrillation, inflammation, histological myocarditis, endomyocardial biopsy, radiofrequency ablation

## Abstract

The aim of the study was to evaluate the inflammatory changes in the myocardium, based on endomyocardial biopsy (EMB) data in patients undergoing radiofrequency ablation (RFA) for idiopathic atrial fibrillation (AF). A total of 67 patients with idiopathic AF were enrolled in the study. Patients underwent the intracardiac examination, RFA of AF, and EMB with histological and immunohistochemical studies. The catheter-treatment effectiveness, and occurrence of early and late recurrences of atrial tachyarrhythmias, were assessed depending on the identified histological changes. Nine patients (13.4%) did not have any histological changes in the myocardium according to EMB. Fibrotic changes were detected in 26 cases (38.8%). Inflammatory changes according to the Dallas criteria were observed in 32 patients (47.8%). The follow-up period for patients averaged 19.3 ± 3.7 months. The effectiveness rates of primary RFA were 88.9% in patients with the intact myocardium, 46.2% in patients with fibrotic changes of varying severity, and 34.4% in patients with the presence of criteria for myocarditis. No early recurrence of arrhythmias was observed in patients with unchanged myocardia. The presence of inflammatory and fibrotic changes in the myocardium increased the rates of early and late arrhythmia recurrences and accordingly halved the effectiveness RFA of AF.

## 1. Introduction

The introduction should briefly place the study in a broad context and highlight atrial fibrillation (AF) as the most common arrhythmia with heterogeneous clinical manifestations often observed in clinical practice. AF is the cause of one-third of hospitalizations for cardiac arrhythmias. Current clinical guidelines recommend avoiding the term “lone” or isolated AF. However, in clinical practice, there are patients without any clinical and echocardiographic signs of cardiovascular and pulmonary disease, and conditions such as acute infections, recent cardiac surgery, thoracic or abdominal operations, and systemic inflammatory diseases [1].

It Is known that an essential element of AF pathophysiology is atrial remodeling, which has three main components: structural, electrical, and mechanical [2]. Inflammation is an important factor in structural remodeling. Indeed, Lau et al. showed that the inflammatory infiltration in the atrial myocardium was detected in patients with isolated AF. Further evidence for the association between AF and inflammation is increased concentrations of serum inflammatory markers, such as C-reactive protein (CRP), TNF-α, interleukins, and cytokines [3]. Besides, the level of serum inflammatory markers increases in patients with both isolated AF and AF associated with the underlying disease [4]. However, it is challenging to explain the occurrence of cellular infiltration and an increase in inflammatory markers only by the fact of AF without the presence of an infectious agent.

The diagnosis of idiopathic AF often suggests the presence of unrecognized myocardial lesion with a certain etiology. One of the most common causes for idiopathic AF is chronic myocarditis occurring without vivid clinical manifestation [5]. Endomyocardial biopsy (EMB)-based confirmation of active myocarditis, which can initially be suspected in patients with a combination of minimal clinical and laboratory-instrumental signs, provides the basis for successful administration of anti-inflammatory and immunosuppressive therapy.

Myocarditis can be difficult to diagnose, primarily due to the heterogeneity of its clinical manifestations. Data on the prevalence of myocarditis are limited, and there are no studies on the subclinical course of this disease. Available studies of myocardial biopsies in young people who suddenly died suggest the presence of myocarditis in 2 to 42% of cases [6,7]. The absence of a specific clinical picture, clear association with the previous infection, changes in exercise tolerance, and changes in ECG and echocardiography data often do not allow suspicion of the presence of myocarditis; spontaneous recovery completely excludes further diagnostic searches in this direction. However, the evolution of viruses, their rapid spread, tendency to chronicity, and the onset of an autoimmune component increase the number of patients with progressive cardiac dilatation and a poor prognosis. Currently, myocarditis is commonly defined as an inflammatory heart disease where diagnosis is established according to the histological (Dallas criteria), immunological, and immunohistochemical criteria (14 or more lymphocytes per square mm, including up to four monocytes and seven or more CD3^+^ T-lymphocytes). The cause of myocarditis often remains unknown. It is believed that the cause of myocardial damage in most cases is a viral infection. Therefore, polymerase chain reaction of the myocardial tissue sample allows detection of enterovirus, adenovirus, parvovirus B19, herpes simplex virus types 1, 2, and 6, cytomegalovirus, and Epstein-Barr virus during histological and immunological studies [8]. Delayed acute viral disease and occurrence as a result of latent subacute myocarditis is probably not uncommon in the modern world, but the occurrence spontaneously resolves without consequences in most patients. In some cases, myocarditis persists and leads to the development of fibrosis, onset of tachyarrhythmias and/or heart failure. AF represents one of these manifestations. Current treatment for AF mainly involves the use of interventional techniques such as radiofrequency isolation of the pulmonary veins, application of multiple damage lines, etc. At the same time, intracardiac intervention allows an EMB to be performed to confirm inflammatory or degenerative myocardial disease. Therefore, we set a goal to evaluate the contribution of inflammation to the clinical outcomes of the radiofrequency ablation (RFA) of AF.

## 2. Materials and Methods

### 2.1. Study Population and Design

We examined 274 patients (182 men, 66.4%) aged 30 to 55 years (mean age of 42.2 ± 18.6 years) admitted to the clinic with a diagnosis of AF. The inclusion criterion was diagnosis of idiopathic AF. The exclusion criterion was the presence of associated diseases: arterial hypertension, obesity, diabetes mellitus, dyslipidemia, cardiovascular, autoimmune, pulmonary diseases, thyroid pathology, or other diseases that could potentially cause AF. While staying in the hospital, all patients underwent the following examinations: 12-lead ECG, Holter ECG monitoring, bicycle ergometry, six-minute walking test, clinical and biochemical blood tests, 24-h blood pressure monitoring, transthoracic echocardiography, stress test to exclude coronary heart disease (myocardial perfusion scintigraphy or stress ECHO), and in case of a positive stress test, multispiral computer or invasive coronary angiography to exclude atherosclerotic changes in the coronary arteries and, accordingly, ischemic genesis of arrhythmia. If pathological changes were detected during the examination, the patient was excluded from the study. The study included 67 patients (22.9%), in whom the cause of AF was not found by any methods available to us. Of these patients, 43 individuals (64.2%) were men aged 34 to 50 years (mean age of 41.1 ± 7.6 years). Upon admission to the clinic, all these 67 patients complained of palpitations. Persistent AF was in 29 patients (43.3%), and long-term persistent (for more than one year) in 38 cases (56.7%). The duration of arrhythmic history was 5.7 ± 1.4 years. The specific pharmacotherapy before admission to the hospital was not carried out, since it was initially believed that all patients had isolated AF. All patients with the persistent AF took antiarrhythmic drugs for AF prevention at the outpatient stage (before the hospitalization): amiodarone in 73.2% of cases, sotalol in 11.3% of patients, and propafenone in the rest of patients (15.6%). For the arrhythmia-paroxysms termination, amiodarone was used as a first line and propafenoneas a second line. All patients with long-term persistent AF before hospitalization were receiving β-blockers as drugs for heart-rate control. However, it should be noted that, according to the past medical history, patients previously took all available antiarrhythmic drugs, and the average number of medications taken was 2.8. At hospital discharge, after the interventional AF treatment, all patients were prescribed with the antiarrhythmic therapy for three months at least. Amiodarone was prescribed to patients who had long-term persistent AF before AF ablation and propafenonein to the remaining patients. The anticoagulation therapy was not prescribed before AF ablation, as all patients had no more than 1 CHA2DS2-VASc score. After the ablation, all patients were with the oral anticoagulation for three months.

The study was approved by the Local Ethics Committee and conformed to the Declaration of Helsinki on Human Research. Written informed consent was obtained from every patient after explanation of the protocol, its aims, and potential risks.

All patients underwent the interventional AF treatment. Computer angiotomography with reconstruction of the left atrium, transesophageal echocardiography, and anticoagulant therapy were used as preparation for the procedure. Also cardiac magnetic resonance (MRI) with contrast and late gadolinium enhancement was performed in all patients before AF ablation with the Vantage Titan 1.5T scanner (Toshiba, Tokyo, Japan). In all cases, before ablation, electroanatomic mapping of the left atrium was performed with the reconstruction of bipolar maps and the identification of areas with reduced amplitude, as well as with an electrical “scar”. The radiofrequency antral isolation of the pulmonary veins, the posterior wall of the left atrium, and the mitral isthmus of the heart were performed using the CARTO system (Biosense Webster, Irvine, CA, USA) in all cases. Vein isolation was monitored with the Lasso circular mapping catheter (Biosense Webster, Irvine, CA, USA). The sinus rhythm, if necessary, was restored by electrical cardioversion. After the sinus rhythm restoration, EMB was performed. Biopsies were taken under X-ray control from the apex, interventricular septum (IVS), and right ventricular outflow tract. Of 67 patients who underwent EMB, 47 also underwent interatrial septum (IAS) biopsy under the transesophageal ultrasound control without any complications. The obtained samples were labeled accordingly and fixed in 10% buffered neutral formalin.

### 2.2. Histological and Immunohistochemical Studies

Paraffin sections were stained with hematoxylin and eosin, picrofuxin, and tolluidine blue; cardiac amyloidosis was excluded in patients over 45 years old by staining with Congo red. An immunohistochemical study was carried out to determine the immunophenotype of infiltrating cells (CD3, CD45, CD68) in each fragment of the endomyocardium and to detect the expression of cardiotropic virus antigens. The following antibodies were used: rabbit polyclonal antibodies to CD3 (Spring BioScience, Pleasanton, CA, USA), mouse monoclonal antibodies to CD45R0 (MONOSAN, Uden, The Netherlands), mouse monoclonal antibodies to CD68 (DCS), rabbit polyclonal antibodies to VP-2 protein of parvovirus B19 (Dako Cytomation), mouse monoclonal antibodies to VP-1 protein of enteroviruses (MONOSAN, Uden, The Netherlands), rabbit polyclonal antibodies to herpes virus type 2 (Dako Cytomation, Glostrup, Denmark), mouse monoclonal antibodies to herpes simplex virus type 1 (Leica Microsistems, Wetzlar, Germany), mouse monoclonal antibodies to adenovirus (Leica Microsistems, Wetzlar, Germany), mouse monoclonal antibodies to early cytomegalovirus nuclear protein (Dako Cytomation, Glostrup, Denmark), and mouse monoclonal antibodies to Epstein-Barr virus LMP antigen (Dako Cytomation, Glostrup, Denmark). High-temperature antigen unmasking was performed when performing studies with antibodies to CD3, CD68, parvovirus B19, adenovirus, cytomegalovirus, herpes simplex virus type 1, 2, and LMP antigen of Epstein-Barr virus. A multivalent horseradish peroxidase diaminobenzidine (HRP-DAB) detection system (Spring BioScience, USA) was used to visualize the studied antigens.

The study of histological preparations was carried out at the light-optical level using an AxioLab A1 Zeiss microscope (Carl Zeiss AG, Jena, Germany). The 1997 Marburg agreement was used for morphological verification of myocarditis [9]. The infiltrating cells were counted, taking into account their immunophenotype (CD3, CD45, and CD68) [10,11].

### 2.3. Clinical Follow-Up

All patients had sinus rhythm on discharge from the clinic. After the procedure, all patients were prescribed antiarrhythmic and anticoagulant drugs for three months. The first three months of follow-up was considered a blind period, and the effect of the procedure was not evaluated; however, the occurrence of AF episodes was considered early relapses. All episodes of AF more than 30 s recorded on ECG or 24-h ECG monitoring, as well as symptomatic paroxysms, were considered an early relapse. Follow-up included an evaluation of complaints, ECG registration biquarterly, and Holter ECG monitoring twice every six months. The results of EMB and immunohistochemical studies were immediately provided to patients, with appropriate recommendations to follow.

### 2.4. Statistical Analysis

Statistical analysis was performed using the Statistica 10.0 software package and MedCalc 13. Continuous variables were expressed as mean ± standard deviation. The Shapiro-Wilk test was used to assess the normality of variable distribution. To assess the differences between the variables, the non-parametric Mann-Whitney test for independent samples was used. The Spearman test was used to estimate the correlation coefficient between quantitative variables. Efficacy analysis was performed using logistic regression analysis. To evaluate the independent predictors of CRT response, forward-stepwise logistic regression analysis was used with an entry criterion of *p* < 0.05 and a removal criterion of *p* > 0.1. Receiver-operating characteristic (ROC) analysis was used to determine the diagnostic efficiency of the methods. Intra- and inter-observer reproducibility was assessed with intraclass correlation coefficients (ICC): *p*-value < 0.05 was considered significant.

## 3. Results

According to EMB results, no histological changes in the myocardium of the right ventricle (RV) were found in nine patients (13.4%). Fibrotic changes in the myocardium were detected in 26 cases (38.8%) including in predominantly perivascular fibrosis in 11 patients (42.3%), small focal fibrosis in eight patients (30.8%), and perimuscular fibrosis in seven patients (26.9%) (Figure 1, Figure 2 and Figure 3).

Inflammatory changes in the myocardium were detected in 32 patients (47.8%), including nine patients (28.1%) with lymphocytic infiltration of less than 14 lymphocytes per mm^2^ (Figure 4 and Figure 5). The data obtained with EMB from RV and IAS were comparable. Inflammatory changes in RV correspond to a similar finding in IAS, while fibrotic changes in RV correspond to the same evidence in IAS. According to the results of immunohistochemical analysis, the virus expression was detected in one of these patients (3.1%). A combination of human herpes simplex virus type 2 and Epstein-Barr was found. No virus expression was detected in the remaining patients.

According to the Dallas criteria, the presence of histological myocarditis was revealed in 23 patients (34.3%) (Figure 6). Moreover, the virus expression was detected in 18 of these patients (78.3%), according to the results of immunohistochemical analysis. One patient (5.6%) was found to express three viruses: enterovirus, human herpes simplex virus type 1, and Epstein-Barr virus; six patients (33.3%) had the presence of two viruses: one patient had a combination of parvovirus and herpes simplex virus type 2; three patients had a combination of enterovirus and herpes simplex virus type 1; and two patients had a combination of Epstein-Barr virus and human herpes simplex virus type 2. The presence of one viral antigen was detected in 11 cases (61.1%), including five patients (27.8%) with Epstein-Barr virus, three patients (16.7%) with enterovirus (Figure 7), two patients (11.1%) with human herpes simplex virus, and one patient (5.6%) with parvovirus. Another five patients (21.7%) did not have viral infection.

When comparing the results of EMB from IAS and electroanatomic mapping, the results of both studies largely coincided. If there was an intact myocardium according to the EMB data, then no areas with reduced amplitude, and no electrical “scar”, were detected according to the bipolar voltage map. If fibrosis was diagnosed by EMB, then according to the bipolar voltage map, areas with reduced amplitude as well as an electrical “scar” were detected (Figure 8).

According to the 16–28-month follow-up results (on average 19.3 ± 3.7 months), the patients were divided into two groups. Group 1 comprised individuals who had no AF recurrence during follow-up according to the objective and subjective examinations. Group 2 comprised patients with reported relapses of AF or other atrial tachyarrhythmias. Data are presented in Table 1.

Considering that 35 patients did not have arrhythmia paroxysms (group 1), the overall efficiency of a single procedure was 52.2%. Early relapses were reported in 26 (41.3%) cases: in 14 patients (22.2%) with fibrotic changes and in 12 patients (19.1%) with inflammatory signs. During further follow-up, the paroxysms of arrhythmias were recorded in 27 patients (42.9%), of whom one patient (1.6%) had an intact myocardium, nine patients (14.3%) had fibrotic changes, and 17 patients (26.9%) had inflammatory signs.

## 4. Discussion

On one hand, the introduction of the RFA AF method into clinical practice, as described by M. Haissaguerre in 1998, opened up the possibility of eliminating arrhythmia. On the other hand, the diagnostic search in many cases was limited by routine ECG registration of tachycardia [12]. The use of endocardial interventions for AF has become globally widespread. However, the overall effectiveness of procedures, according to different authors, rarely exceeds 80% [13]. Changing techniques, and creating additional lines and areas of damage, can increase the efficiency, but to a rather moderate degree. Most likely, the limitations are not caused by a low efficiency of procedure itself, but rather they are due to the fact that a purely mechanistic-anatomical approach is used, as a rule, during the interventional treatment when the veins are electrically isolated and lines are applied without paying attention to the causes of AF. However, the etiology and pathophysiology of AF are multifaceted, and subclinical inflammation, and its consequences including the development of fibrotic changes, may play an essential role in AF development.

In our study, using standard examination methods only in 67 (24.5%) out of 274 patients admitted to the clinic with a diagnosis of idiopathic AF we were unable to detect cardiovascular or other diseases that could potentially explain the onset of arrhythmias. Among these 67 patients with idiopathic AF with no clinical and anamnestic data for the presence of inflammation, almost half (47.8%) of them had the inflammatory changes in the myocardium with the cellular infiltration or criteria for histological myocarditis. Immunohistochemical study allowed detection of the virus expression in the myocardium of most of these patients (59.4%). It should be noted, however, that the lack of data for the presence of a viral infection in negative patients does not exclude the potential presence of other viruses, which could not be detected by the virus-specific diagnostic kits used in our study.

The role of inflammation in the pathogenesis of isolated AF remains equivocal and limited. It is well known that the information regarding the presence of myocarditis and fibrosis could be obtained non-invasively using MRI. In our study, MRI performed in all patients did not disclose any finding consistent with biopsy results. The controversial nature of our data may be explained by the fact that this type of examination is highly dependent on the type of scanner and the doctor who conducts the study. Another important method in the non-invasive diagnostics of inflammation is the assessment of systemic concentrations of inflammatory biomarkers. In several studies, the association of inflammation markers and AF occurrence has been demonstrated. In 2018, we published an article in which we showed that in patients with isolated AF, the plasma levels of TNF-a, IL-1ß, IL-6, IL-8, neopterin, and high-sensitivity C-reactive protein exceeded that in comparison with healthy volunteers, while the concentration of IL-10 did not differ. Markers of renin-angiotensin-aldosterone system, particularly plasma renin activity and aldosterone concentration, were within the range of reference values. In this study, a specific serum marker of the latent myocarditis in patients with AF was IL-6 at a concentration of more than 1.6 pg/mL, and the marker of latent viral myocardial infection was neopterin at concentrations >13.2 nmol/L [14]. The increased levels of these markers can serve as a sign of latent viral myocarditis in AF of unclear etiology.

While assessing the effectiveness of the intervention based on the results of histological examination, we found that the effectiveness of primary RFA in patients with the intact myocardium was 88.9%. However, the effectiveness of primary RFA dropped to 46.2% in patients with fibrotic changes of varying severity and was only 34.4% in the presence of criteria for histological myocarditis. Early recurrences of arrhythmias were absent in patients with unchanged myocardium. Patients with the presence of fibrotic changes more often (53.8%) had early relapses and less often late relapses (34.6%), which, to some extent, can be considered associated with a favorable prognosis, despite the presence of fibrotic changes in the myocardium. We observed the inverse relationship in patients with the presence of inflammatory changes: late relapses were detected more often (53.1%) whereas early relapses were found less often (37.5%). This observation most likely indicates the presence of a persisting and ongoing inflammatory process underlying the onset of recurrent arrhythmias. This portion of patients requires more thorough diagnostics, monitoring, and specific treatment.

Unfortunately, the amount of data on available approaches to diagnose atrial myocardial inflammation in vivo is limited to date. Atrial EMB may be dangerous due to the risk of potential complications, but other diagnostic modalities are not always justified in otherwise absolutely-healthy patients, primarily because it is hard to suspect myocarditis if AF is the only symptom of disease. On the one hand, inflammatory changes occurring in the atria are less dangerous and cannot be the primary cause of sudden cardiac death or heart failure. On the other hand, the capabilities of detecting atrial inflammation are limited. Meanwhile, the atria in comparison with the ventricles are more vulnerable to fibrosis and connective-tissue proliferation due to the lower myocardial mass, which ultimately results in the anisotropic propagation of excitation and the occurrence of atrial tachyarrhythmias.

However, according to the published data, Yamaguchi et al. performed the intracardiac ECHO-guided endocardial biopsy in patients with AF. The authors have shown that biopsy from the right atrium (RA) septum seems to be a feasible and safe technique, although the significance of the RA biopsy in clinical practice is still unclear. Also Yamaguchi et al. detected an inverse relationship between bipolar voltage and fibrosis. However, there was variation in fibrosis, especially in patients whose voltage was in the middle range. At the end of the article, the authors concluded that factors other than fibrosis could affect voltage, e.g., myocyte cell size, myocyte disarray, intercellular-spacing, myofibrillar loss, infiltration with adipocytes. However, the impacts of these factors on voltage have not been analyzed, and future studies are warranted [15]. In our study, we conducted endocardial biopsies from IAS with the transesophageal ECHO control. We agree that this procedure seems to be a feasible and safe technique, however, when comparing the results of EMB from IAS and electroanatomic mapping, in our case the results of both studies have largely coincided. Thus, the significance of the RA biopsy in clinical practice requires additional justification. Moreover, it is well known that Mitrofanova et al. in the article “Histological evidence of inflammatory reaction associated with fibrosis in the atrial and ventricular walls in a case-control study of patients with history of atrial fibrillation” have shown that histological signs of chronic inflammation affecting ventricular myocardium are strongly associated with AF and demonstrate significant correlation with fibrosis extent that cannot be explained by cardiovascular comorbidities otherwise [16].

The inflammatory changes in the atrial myocardium identified in our study raise more questions than answers. First of all, this work using an EMB-based approach represents an investigative study and should not be extended to widespread clinical practice. However, the identified inflammatory changes in the atrial myocardium in AF patients require further comprehensive investigation and, above all, the search for the ways of non-invasive or minimally invasive diagnostics. On the other hand, the implementation of such methods in clinical practice would require baseline data, which prompt a clinician to suspect the presence of myocarditis.

The second question is whether the presence of subclinical inflammation and viral infection in the cardiomyocytes of otherwise healthy patients requires therapy. If so, what should be the treatment goal: the elimination of arrhythmia as the leading symptom or the virus elimination? If the arrhythmia is eliminated, will this mean that the myocardium is healthy and there is no viral infection and subacute myocarditis, which could lead to sudden cardiac death or the development of inflammatory cardiomyopathy and heart failure after an indefinite time? If the treatment goal is to eliminate the virus, then comprehensive etiotropic therapy, which, as a rule, takes longer than one month, would be a priority without a doubt. However, it may be challenging to convince otherwise-healthy patients of treatment necessity. The most important question is how to monitor the effectiveness of the therapy: via the second biopsy or other diagnostic modalities? Will these methods allow study of the processes occurring in the myocardium? The infection consequences, i.e. the amount and extent of fibrous tissue in the atria and ventricles, remains an essential problem as they are known to lead to the onset and progression of atrial and ventricular remodeling.

Only nine (13.4%) of a rather small group of patients included in the study had no histological changes. These cases perhaps may be considered a variant of true electrical heart disease manifested in the form of AF. However, this statement is limited by the capabilities of diagnostic methods used in the study.

### 4.1. Study Limitations

The present study had some limitations. Our study was single-center and included a small group of patients with isolated AF. Because of this, this study is underpowered and its findings are only hypothesis-generating. Therefore, future research requires larger scale multicenter studies. In our work, we did not assess the adverse cardiac events such as thromboembolic events or heart failure during follow-up. Moreover, we used RFA as a treatment for AF, but we did not treat underlying histological myocarditis. We did not perform the high-density mapping using a multi-pole diagnostic catheter for bipolar voltage maps of the left atrium, but performed mapping with an ablation catheter.

### 4.2. New Knowledge Gained

The significance of EMB is supported by its ability to reveal the etiology of histological myocarditis and AF, specifically, in MRI negative patients. EMB results confirm the presence of histological myocarditis and, accordingly, may help in choosing etiotropic treatment.

## 5. Conclusions

The diagnostic term “idiopathic AF” is used unreasonably often in clinical practice. According to our data obtained during a standard examination, only 24.5% of patients had no diseases, which could potentially lead to the development of arrhythmia. The histological findings showed that only about 10% of AF patients had a true idiopathic form of arrhythmia, while half of AF patients had latent inflammatory changes in the myocardium and the remaining patients had fibrotic changes as a result of inflammation. Our findings indicate that the presence of inflammatory and fibrotic changes in the myocardium may increase the rates of early- and late-arrhythmia recurrences in patients undergoing RFA for AF. However, further studies are warranted to investigate in-depth this possible relationship.

## Figures and Tables

**Figure 1 jcm-12-01254-f001:**
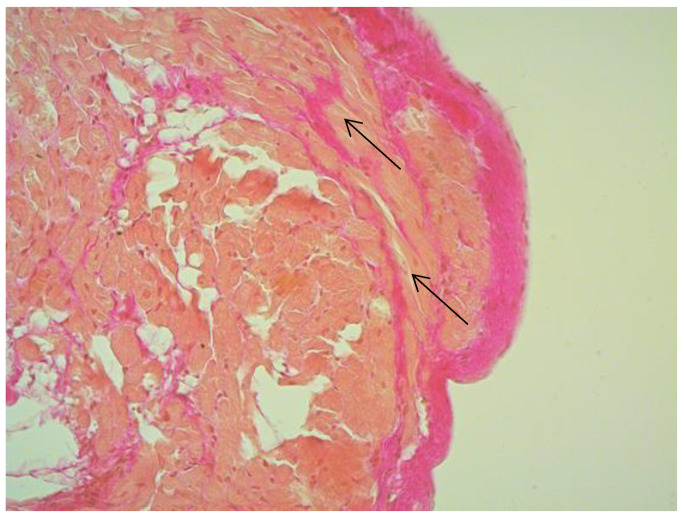
Perimuscular fibrosis in IVS, ×100, staining according to Van Gieson. The arrows indicate the connective tissue proliferation.

**Figure 2 jcm-12-01254-f002:**
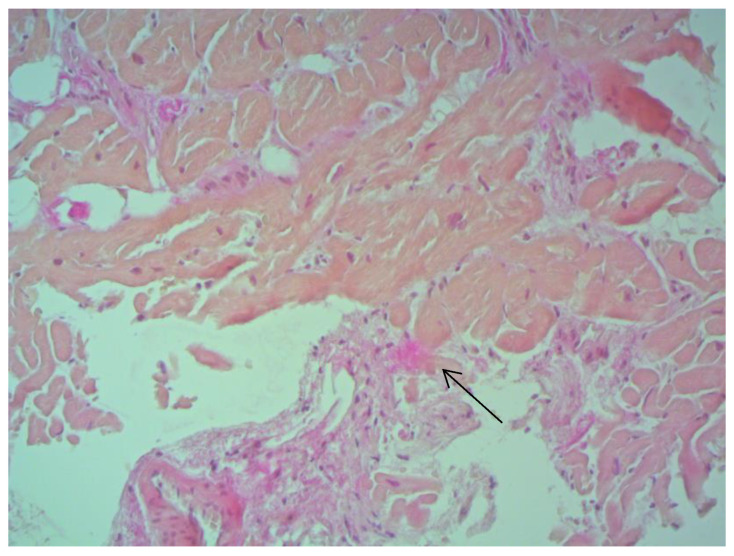
Small focal fibrosis in IVS ×200, staining according to Van Gieson. The arrow indicates the focus of connective tissue proliferation.

**Figure 3 jcm-12-01254-f003:**
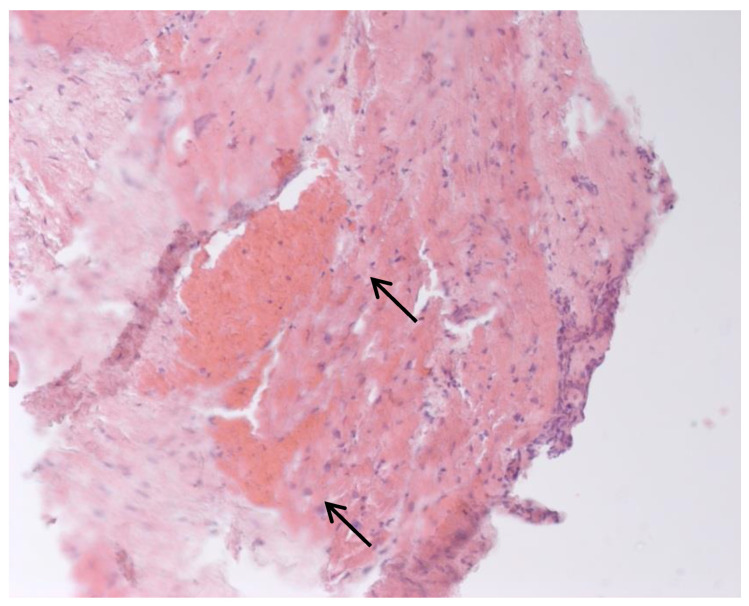
Fibrosis in IAS ×200, staining according to Van Gieson. The arrows indicate the foci of connective tissue proliferation.

**Figure 4 jcm-12-01254-f004:**
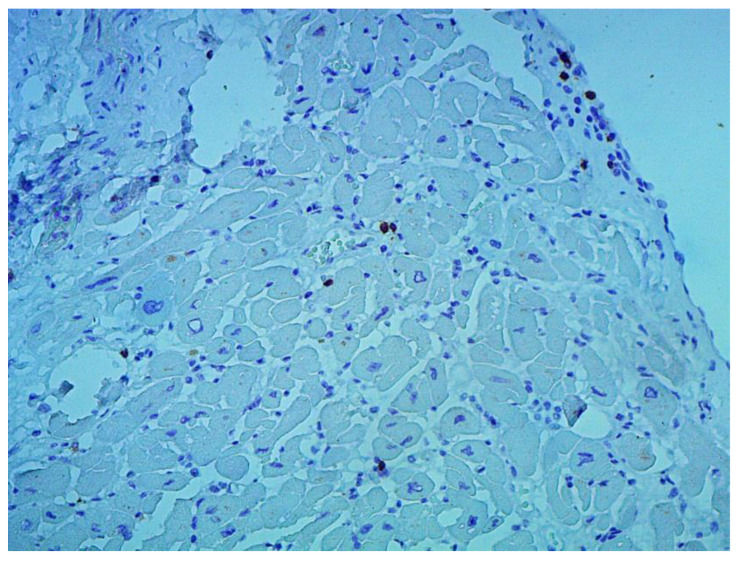
Endomyocardial infiltration with CD3+ lymphocytes. Immunohistochemical study, antibodies to the Epstein-Barr virus, ×200.

**Figure 5 jcm-12-01254-f005:**
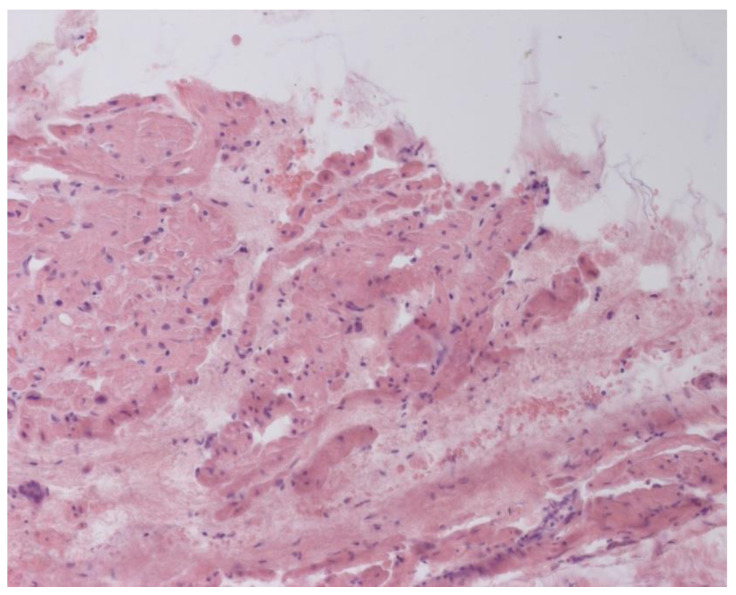
Endomyocardial infiltration of IAS with lymphocytes, ×200. Hematoxylin-Eosin Staining.

**Figure 6 jcm-12-01254-f006:**
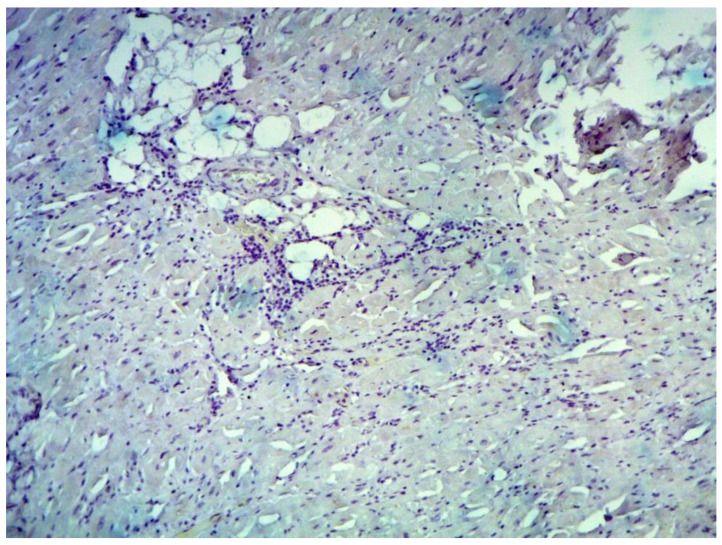
Active lymphocytic histological myocarditis, ×100. Hematoxylin-Eosin staining.

**Figure 7 jcm-12-01254-f007:**
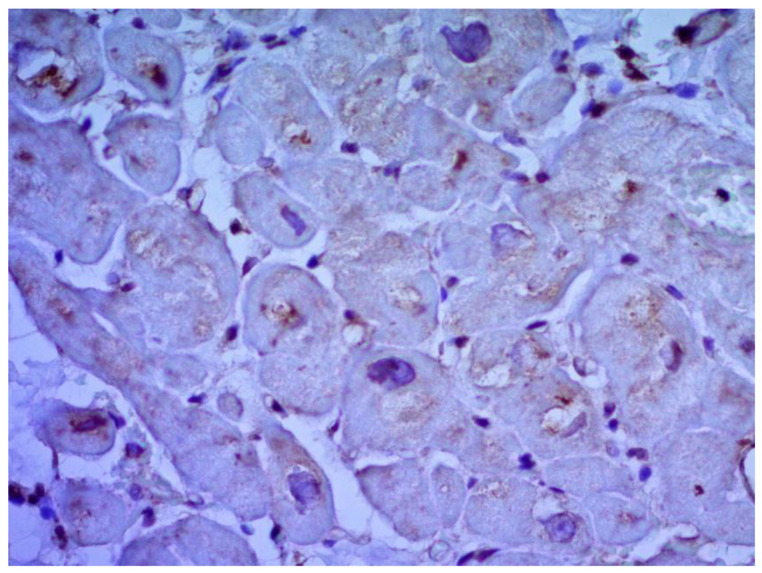
Expression of the enterovirus antigen VP1 in the myocardium. Immunohistochemical study, monoclonal mouse antibodies, ×400.

**Figure 8 jcm-12-01254-f008:**
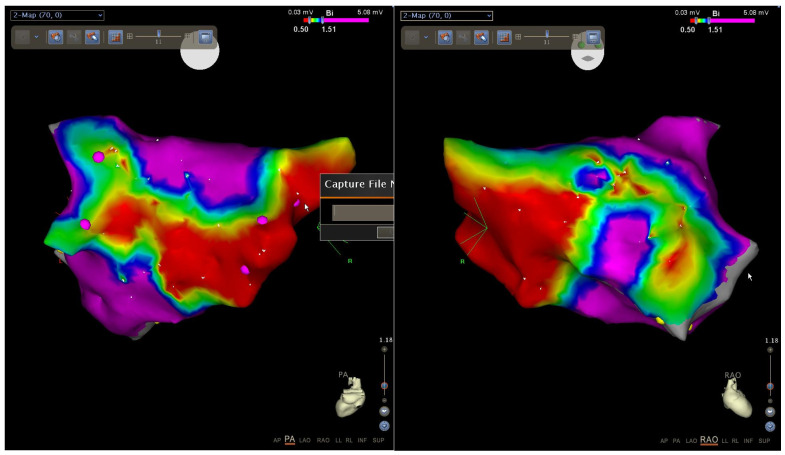
Bipolar voltage map of a patient with the ineffective RFA of AF and fibrosis identified by IAS biopsy. Posterior and right oblique projections. Notes: red-color areas of the atrial signal amplitude less than 0.5 V; magenta areas of the atrial signal amplitude more than 1.5 V; gradient color signals - a transition zone. Purple dots indicate the PV ostiums.

**Table 1 jcm-12-01254-t001:** Effectiveness of RFA AF in various histological changes in the myocardium.

Parameters	Group 1 (*n* = 35), *n* (%)	Group 2 (*n* = 32)	
		Early Recurrence, *n* (%)	Late Recurrence, *n* (%)
No changes in EMB (*n* = 9)	8 (88.9)	-	1 (11.1)
Fibrotic changes (*n* = 26)	12 (46.2)	14 (53.8)	9 (34.6)
perivascular fibrosis (*n* = 11)	3 (11.5)	8 (30.8)	7 (26.9)
small focal fibrosis (*n* = 8)	3 (11.5)	5 (19.2)	2 (7.7)
perimuscular fibrosis (*n* = 7)	6 (23.1)	1 (3.8)	-
Inflammatory changes (*n* = 32)	15 (46.9)	12 (37.5)	17 (53.1)
lymphoid infiltration (*n* = 9)	4 (12.5)	3 (9.5)	5 (15.6)
histological myocarditis (*n* = 23)	11 (34.4)	8 (25.0)	12 (37.5)

## Data Availability

According to the internal regulations of the Institute, all data are the property of the Institute and can only be provided anonymously after an official request.

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
