# Peer review of "Isolated Atrial Fibrillation, Inflammation and Efficacy of Radiofrequency Ablation: Preliminary Insights Based on a Single-Center Endomyocardial Biopsy Study"

_jcm, 2023, doi:10.3390/jcm12041254_

Round 1
Reviewer 1 Report
In the article entitled "Lone Atrial Fibrillation, Inflammation and Efficacy of Radiofrequency Ablation", Batalov et al. evaluated the inflammatory changes in the myocardium based on endomyocardial biopsy (EMB) data in patients undergoing radiofrequency ablation (RFA) for idiopathic atrial fibrillation (AF).
Among 67 patients enrolled in the current study, they found that nine patients (13.4%) did not show any histological changes in the myocardium according 17 to EMB. Fibrotic changes were detected in 26 cases (38.8%), while inflammatory changes according to the Dallas criteria were observed in 32 patients (47.8%). The effectiveness rates of primary RFA were 88.9% in patients with the intact myocardium, 46.2% in patients with fibrotic changes of varying severity, and 34.4% in patients with the presence of criteria for myocarditis. No early recurrence of arrhythmias was observed in patients with unchanged myocardium.
Major comments:
- As also the authors suggest, the term "lone" atrial fibrillation should be avoided according to European guidelines. Please change the title with "isolated". In this regard, please discuss the following study: Wyse DG, Van Gelder IC, Ellinor PT, Go AS, Kalman JM, Narayan SM, Nattel S, Schotten U, Rienstra M. Lone atrial fibrillation: does it exist? J Am Coll Cardiol. 2014 May 6;63(17):1715-23. doi: 10.1016/j.jacc.2014.01.023. Epub 2014 Feb 12. PMID: 24530673; PMCID: PMC4008692.
- Methods: The authors state: "We examined 274 patients (182 men (66.4%)) aged 30 to 55 years (mean age of 42.2 ± 85 18.6 years) admitted to the clinic with a diagnosis of idiopathic AF." Then they state that all patients underwent coronary angiography to exclude atherosclerotic changes in the coronary arteries. So all patients with idiopathic AF underwent coronary angiography just for screening ever if it was not clinically indicated?
- Methods: After all the exams, the study included 67 patients (22.9%), in whom the cause of AF was not identifiable. So I assume that in the others (77.1%), even if they were thought to be "idiopathic", AF was related to something else. Correct? If so, please change the statement that 274 patients with "idiopathic" AF were initially screened. 77.1% of non-idiopathic is a very wide rate.
- What about voltage maps? Do these maps correlate with the fibrotic areas that the investigators found during mapping? If all patients underwent RF ablations (the authors state CARTO + Lasso), maps will be surely available. Please examine those maps and provide details. This will surely enhance the message of this paper.
- Why did the authors collect samples from the apex, interventricular septum (IVS), and right ventricular outflow tract? What about eventual scar zones in the LA?
- I would avoid the use of the term "myocarditis", it should be replaced with something like "inflammation". Myocarditis is more a clinical events, along with myocardial inflammation and in this case, clinical features were lacking.
Minor comments:
- Please review the manuscript for typos and language errors such as "palpitation>palpitations" ecc.
Author Response
Thanks for the review, answers to questions in the attached file
Reviewer 2 Report
In the reviewed manuscript, Batalov et al. aimed to assess the inflammatory changes in the left ventricular muscle based on the endomyocardial biopsy in patient undergoing ablation for idiopathic atrial fibrillation. The text is correctly written. However, I have the following important comments:
1. I don't think that it is ethically justified to perform the endomyocardial biopsy in the study population. The information regarding the presence of myocarditis and fibrosis could be obtained non-invasively using cardiac magnetic resonance. In my opinion, the study participants were unnecessary exposed at the risk of the endomyocardial biopsy.
3. The authors evaluated tissue samples obtain from the right ventricular muscle, but not from the atria. In fact, it is unclear whether atrial tissues were involved.
4. The study is underpowered . In fact, the authors did not perform any statistical comparsion among the subgroups (Table 1) and the study conclusion (The presence of inflammatory and fibrotic changes in the myocardium increased the rates of early and late arrhythmia recurrences and accordingly halved the effectiveness RFA of AF.) is not supported by the presented data.
5. The authors did not assessed systemic concentrations of inflammatory biomarkers, e.g. CRP.
6. Pharmacotherapy of the study participants is not described in the manuscript.
7. Figures 1 and 2 lack arrows. However, the authors state that the arrows indicate the connective tissue proliferation.
8. The references should be formatted according to the journal style.
Author Response

(The authors gave the same response as above.)

Reviewer 3 Report
The authors evaluated the relationship between inflammatory and fibrotic changes in the myocardium and recurrence after atrial fibrillation ablation. I read this paper with interest and care. There are serious concerns and inconsistencies in this study.
# The authors performed the biopsy in the right ventricles. As indicated in the reference, atrial fibrillation has been proven to be associated with inflammation of the myocardium by biopsy of the atria. The method of biopsy the ventricles does not make sense in atrial fibrillation. This is the biggest limitation of this study
# “There have been no available approaches to diagnose atrial myocardial inflammation in vivo to date” in Discussion section. This is a wrong and misleading statement! Yamaguchi T et al. performed atrial biopsy in patients with atrial fibrillation (Atrial Structural Remodeling in Patients With Atrial Fibrillation Is a Diffuse Fibrotic Process: Evidence From High-Density Voltage Mapping and Atrial Biopsy. J Am Heart Assoc. 2022).
# What about voltage mapping in atrium during atrial fibrillation ablation?
# Many patients with persistent or long-standing persistent atrial fibrillation have no symptoms. However, the authors state that all patients complained of palpitation in Methods section. It is surprising. This differs from numerous previous reports.
Author Response

(The authors gave the same response as above.)

Reviewer 4 Report
The work presented by the group of Batalov et al. appears to be well structured. The idea of assessing inflammatory changes in the myocardium based on endomyocardial biopsy data in patients undergoing radiofrequency ablation for idiopathic atrial fibrillation is intriguing and interesting.
In addition, several questions emerge from reading the article.
1. Although endomyocardial biopsy is more specific it seems to be limited to tissue segments. Therefore, is there a possibility of subjecting patients to cardiac MRI with contrast to assess for signs of myocarditis and late gadolinium enhancement? Has the possibility of performing an MRI using software described in the literature, useful for quantifying the degree of atrial fibrosis, been evaluated?
3. What echocardiographic imaging data were collected before the ablation procedure? Was atrial strain considered for indirect assessment of atrial fibrosis?
Some observations:
Since biopsy specimens are exclusively taken from the ventricle, it would be useful to supplement the manuscript with data from a review of the literature on the possible correlation between the presence of inflammatory damage/fibrosis in the ventricle and in the atrium.
It might be useful to perform electroanatomic mapping in patients undergoing RFA and those with recurrence in order to understand the electroanatomic mechanisms of AF recurrence.
Follow-up with MRI of patients who presented fibrosis and myocarditis might be useful to assess their evolution.
Author Response

(The authors gave the same response as above.)

Round 2
Reviewer 2 Report
The authors improved the manuscript, but they did not addressed all comments adequately.
I have the following comments when assessing the revised manuscript:
My initial comment 1: I don't think that it is ethically justified to perform the endomyocardial biopsy in the study population. The information regarding the presence of myocarditis and fibrosis could be obtained non-invasively using cardiac magnetic resonance. In my opinion, the study participants were unnecessary exposed at the risk of the endomyocardial biopsy.
Author's answer: The study was conducted in accordance with the Declaration of Helsinki, and approved by the Institutional Ethics Committee of Cardiology Research Institute, Tomsk National Research Medical Center, Russian Academy of Science (protocol code 163 and 08/11/2017). Informed consent was obtained from all subjects involved in the study. In our study, we performed an MRI with contrast and late gadolinium enhancement in all patients, but we did not find a significant correlation between the MRI data and biopsy results. We used the Vantage Titan 1.5T scanner (Toshiba). But, in our opinion, this type of examination is highly dependent on the type of scanner and the doctor who conducts the study. We have not mentioned the MRI data we found in the article due to their controversial nature. The significance of EMB is supported by its ability to reveal the etiology of histological myocarditis and AF, specifically, in MRI negative patients. EMB results confirm the presence of histological myocarditis and, accordingly, may help in choosing etiotropic treatment.
My recommendation: Please comment on the limitations of MRI and performed MRI examinations in the study participants in the revised manuscript.
My initial comment 2: The authors evaluated tissue samples obtain from the right ventricular muscle, but not from the atria. In fact, it is unclear whether atrial tissues were involved.
My comment on the revised manuscript and author's reply: I am satisfied with the answer and improvements made by the authors.
My initial comment 3: The study is underpowered. In fact, the authors did not perform any statistical comparison among the subgroups (Table 1) and the study conclusion (The presence of inflammatory and fibrotic changes in the myocardium increased the rates of early and late arrhythmia recurrences and accordingly halved the effectiveness RFA of AF.) is not supported by the presented data.
Author's answer: Unfortunately, the number of patients in general and in each group in particular is small enough for statistical significance. In 4.1. Study Limitations we have noted that our study was single-center and included small group of patients with isolated AF.
My recommendation: This is a key issue. In the initial manuscripts conclusions were not supported by the data. I suggest further changes in the manuscript. I recommend: i) do add the study title ": preliminary insights based on a single-center endomyocardial biopsy study"; ii) change the conclusion "The presence of inflammatory and fibrotic changes in the myocardium increased the rates of early and late arrhythmia recurrences and accordingly halved the effectiveness RFA of AF" for "Our findings indicate that the presence of inflammatory and fibrotic changes in the myocardium may increase the rates of early and late arrhythmia recurrences in patients undergoing RFA for AF. However, further studies are warranted to investigate in-depth this possible relationship.", and ii) clearly state in the study limitations section that this study is underpowered and its findings are only hypothesis-generating.
My initial comment 4: The authors did not assessed systemic concentrations of inflammatory biomarkers, e.g. CRP.
Author's answer: On this topic we have published another article: Svetlana Ivanovna Sazonova, Julia Nikolaevna Ilushenkova, Roman Efimovich Batalov, Anna Mihaylovna Gusakova, Julia Vladimirovna Saranchina, Julia Viktorovna Rogovskaya, Sergey Valentinovich Popov. Plasma markers of myocardial inflammation at isolated atrial fibrillation / J Arrhythm. 2018 Jun 26;34(5):493-500. doi: 10.1002/joa3.12083.
My recommendation: Please discuss this issue in the revised discussion.
My initial comment 5: Pharmacotherapy of the study participants is not described in the manuscript.
Author's answer: Specific pharmacotherapy was not carried out, since it was initially believed that all patients had isolated AF, some patients were prescribed antiarrhythmic therapy with class 1c and class 3 drugs. The arrhythmia paroxysms termination was carried out according to current recommendations.
My recommendation: Please discuss pharmacotherapy of the study participants in the revised methods section.
My initial comment 6: Figures 1 and 2 lack arrows. However, the authors state that the arrows indicate the connective tissue proliferation.
Author's answer: This error has been corrected.
My comment on the revised manuscript and author's reply: I am satisfied with improvements made by the authors.
My initial comment 7: The references should be formatted according to the journal style.
Author's answer: The references have been checked.
My comment on the revised manuscript and author's reply: I am satisfied with improvements made by the authors.
Author Response
Thanks for the review.

Reviewer 3 Report
This manuscript has been improved.
However, it needs some corrections.
# The figure of the voltage map (Figure 8) needs to be changed. The figure shown is after pulmonary vein isolation and posterior wall isolation. The figure before ablation needs to be shown. Also show the number of mapping points. Which group does the figure shown belong? White dots are not LV, are they PV ostium?
# “The data obtained with EMB from RV and IAS was comparable” (page 4, line 179).. In order to maintain objectivity as to whether these are true or not, correlations (r) etc. must be used.
Author Response
Thanks for the review.
